# Associations between Neighborhood Deprivation Index, Parent Perceptions and Preschooler Lifestyle Behaviors

**DOI:** 10.3390/children8110959

**Published:** 2021-10-24

**Authors:** Carolina Bassul, Clare A. Corish, John M. Kearney

**Affiliations:** 1School of Biological and Health Sciences, City Campus, Technological University Dublin, Kevin Street, D08 TKF7 Dublin, Ireland; 2School of Public Health, Physiotherapy and Sports Science, University College Dublin, Belfield, D04 V1W8 Dublin, Ireland; clare.corish@ucd.ie

**Keywords:** neighborhood environment, children dietary intake, children physical activity, TV screen time, deprivation index, parents’ perceptions, pre-school children

## Abstract

Parental perceptions and use of neighborhood facilities are important factors that are related to children’s dietary intake and physical activity. The aim of this study was to examine the association between neighborhood deprivation index, parents’ perceptions of their neighborhood environment, and healthy/unhealthy markers of child dietary intake, physical activity, and TV screen time. This cross-sectional study was conducted in Dublin, Ireland. The lifestyle behaviors among children and parental perceptions of their neighborhood environment were reported by the parents of 276 children aged 3–5 years by using parent-completed questionnaires. Deprivation index was assessed using the geographic information system (GIS). Data were analyzed using binary logistic regression, adjusting for socio-demographic confounders. In adjusted models, high deprivation index was associated with parental perception of the neighborhood as unsafe for walking and cycling due to crime (OR 1.59, 95% CI 1.04–2.43, *p =* 0.031) and children’s low engagement in structured physical activity (OR 0.35, 95% CI 0.17–0.72, *p =* 0.004). Parental perceptions of an unsafe neighborhood due to heavy traffic were negatively correlated with children’s active play (OR 0.73, 95% CI 0.55–0.95, *p* = 0.022). Children whose parents reported high satisfaction with the number of local sit-in and takeaway restaurants were 41% more likely to consume confectionary/sugar sweetened beverages (SSBs) weekly. In this age group, parents play an important role in children’s lifestyle behaviors; therefore, a better understanding of parents’ perceptions and their use of neighborhood facilities could contribute to creating a healthy environment for this age group.

## 1. Introduction

It is widely acknowledged that healthy lifestyle habits acquired in early childhood extend into adolescence and adulthood [1,2]. The role of social and physical environments in determining children’s food consumption and physical activity levels is as important as individual behaviors [3]. Therefore, the neighborhood environment has recently received greater attention from policymakers and researchers [4,5]. For children, the neighborhood environment includes the food outlet(s) where their families purchase food, the streets and roadways on which they travel, the parks and playgrounds in which they spend leisure time, and the other children and adults with whom they interact [6]. Therefore, parent interactions with other residents, as well as the characteristics of a neighborhood, can influence perceptions of the neighborhood environment, both positively and negatively [7]. These perceptions are linked with psychosocial variables such as social support, social cohesion, and trust, as well as with other neighborhood factors such as a sense of safety (e.g., crime and traffic) and socioeconomic status [7]. These characteristics of the neighborhood environment may affect lifestyle behaviors, weight status, and general health of children [6,8].

Based on their perceptions about neighborhood food and physical activity environment, parents usually decide whether their child is allowed to play outside, walk or cycle to school, use neighborhood recreational facilities, and where to purchase foods [9]. For example, parental concerns about traffic safety and crime were associated with less time playing outdoors in primary school children [10,11,12]. A meta-analysis reported that high levels of crime were associated with a reduction in children’s physical activity by 0.13 h per week [13]. Another study reported a positive association between parents’ perceptions about greater availability and better condition of playgrounds and children’s higher physical activity during leisure time [9] while a US study showed that greater parental satisfaction with the food shopping environment was associated with higher consumption of fruits and vegetables among 3–7 year-olds [14]. Furthermore, parents’ satisfaction with the local food environment may affect home food availability and, consequently, what they feed their children [15,16,17]. A study with 3670 children and their caregivers demonstrated that caregivers who perceived high availability of healthy foods in the neighborhood were more likely to have more healthy foods available at home [15]. In turn, home food availability has been consistently associated with children’s dietary intake in previous studies [16,18,19].

A neighborhood deprivation index has been used in health research to link health outcomes and economic characteristics in a population [20]. In Ireland, a deprivation index has been developed to provide detail at street level, showing the extent to which every neighborhood, suburb, and village in the state is affluent or deprived [21]. The measurements investigated 10 key indicators from the 2006, 2011, and 2016 censuses and included, for example, the proportion of skilled professionals, education levels, employment levels, and single-parent households found in an area [21]. Studies have shown that children living in highly deprived areas are more likely to have unhealthy dietary behaviors, higher screen time, and, consequently, to be at higher risk of overweight or obesity compared to children living in areas of low deprivation [22].

In Ireland, the longitudinal “Growing up in Ireland” (GUI) study reported that children from more deprived areas participated to a greater extent in unstructured physical activity and screen time while children from areas of less deprivation engaged in structured physical activity (e.g., sports clubs) on a regular basis [23]. Furthermore, children from more deprived areas consumed, on average, 23% more calories daily compared to those from less deprived areas [23]. Similarly, a large international study that included children from the UK showed that obesity prevalence in 4–5 year old children living in the most deprived areas was more than double that of those living in the least deprived areas [24]. A further study from New Zealand investigated the association between neighborhood deprivation, unhealthy food outlets, unhealthy dietary behaviors, and children’s weight status. The authors concluded that the neighborhood availability of unhealthy food outlets was not associated with children’s weight status; however, high deprivation index and unhealthy dietary intake were positively associated with higher body weight in children [22].

Understanding the individual aspects of the neighborhood environment associated with the diet and physical activity/sedentary behaviors of children could have significant positive health effects from a public health perspective. Increased knowledge of the influence of the neighborhood deprivation index, which has demonstrated strong correlations with a range of health and social outcomes measures [22,25,26], and parental perceptions of their local neighborhood on pre-school children’s dietary intakes, physical activity, and screen time behaviors can inform the creation of a healthy environment for this age group [27]. Therefore, the aim of this study was to examine the neighborhood deprivation index and to investigate whether parental perceptions of their neighborhood environment were associated with their pre-school children’s physical activity, TV screen time, and healthy/unhealthy markers of dietary intake.

## 2. Methods

### 2.1. Study Participants and Data Collection Procedures

The Pre-schoolers Health Study is a cross-sectional study comprising 332 child-parent dyads recruited from 25 pre-schools between September 2016 and June 2017. The pre-schools were recruited by using stratified random sampling in different neighborhoods of Dublin (nine preschools were from high deprived areas, eight from medium deprived, and eight from low deprived areas). The inclusion criteria for pre-schools were as follows: provide a structured service for a minimum of 3 h daily and be registered with the state agency responsible for inspecting early years services including pre-schools and similar services for children aged 0–6 years. All children aged 3–5 years attending the pre-school who were free from any medical condition that affects their growth and development and their parents were eligible for the present study (*n* = 670). In the case of siblings, only one child per parent participated. Parents that were not proficient in written English were excluded, as they were not in a position to complete the study questionnaire. Parents were first informed about the study by information leaflet, and they were then invited to participate using a “pack” that included an information letter, consent forms (study consent form and anthropometric measurement consent form), and one parent-completion questionnaire. After one week, a reminder was sent to the parents who did not return the completed questionnaire and a further week was given [16]. A total of 276 parents who completed the questionnaire in full, provided their home address, and participated in the present study, which was approved by the Ethics Committee of the Technological University Dublin (TUD) (Ref 15–109).

### 2.2. Data Collection Instruments

#### 2.2.1. The Pre-Schooler Health Study Questionnaire

The demographic characteristics recorded included child and parent gender and age; the parents self-reported their own height and weight, education level, profession, and marital status.

Children’s lifestyle behaviors such as dietary intake, active play, structured physical activity, and TV screen time were the outcome variables in the present study. In the questionnaire, child dietary intake was assessed using indicators of a healthy (high fruit and vegetable intakes) or unhealthy (high confectionary/sugar-sweetened beverage (SSB) intakes) diet [28,29]. Fruit and vegetable intakes were re-coded as (0) < 1 portion a day or (1) ≥ 1 portion a day, and they were assessed separately [3,16,28]. Confectionary/SSB consumption was re-coded as (0) < 1 time per week or (1) ≥ 1 time per week, following the Irish food dietary guideline recommendations for 1 to 5 year old children [16,30]. Children’s active play and TV screen time were dichotomized as (0) < 1 h daily or (1) ≥ 1 h daily [31]. Structured physical activity was assessed with the question “*Does your child attend any organized physical activity during the week?”* Parents responded “yes” or “no” to this question.

Three questions from the validated ALPHA Environmental Questionnaire [32] were used to assess parental perceptions of their local neighborhood walking and cycling environment*: (i) walking and cycling is unsafe due to heavy traffic*; *(ii) walking and cycling is unsafe due to crime*; *(iii) my local neighborhood is pleasant for walking and cycling*. Response options (and coding) were as follows: strongly disagree (1); disagree (2); neither disagree nor agree (3); agree (4); and strongly agree (5). A further six questions from the standardized Home Environment Interview (HEI) questionnaire [33] were used to assess parental satisfaction with their local neighborhood food outlet environment (e.g., “*how satisfied are you with the number and quality of restaurants?*”), physical activity environment (e.g., *“how satisfied are you with the physical activity facilities?”*), and the suitability of the neighborhood as a place to live and raise children. For each item, a 5-point Likert scale was used: (1) strongly dissatisfied; (2) dissatisfied; (3) neither; (4) satisfied; and (5) strongly satisfied. Parents also reported their child’s usage of their local neighborhood physical activity facilities in response to the following question: “*how often does your child use the neighborhood facilities such as parks and playgrounds?”* The possible responses ranged from (1) no such facilities in my neighborhood; (2) rarely; (3) once a month; (4) a few times a month; (5) once a week; to (6) a few times a week to daily. These were then re-coded as 1–4 ≤ 1 time per week and 5–7 ≥ 1 time per week.

#### 2.2.2. Neighborhood Deprivation Index

Participant addresses were initially verified for accuracy using Google Maps. A CSV file was then created containing a column for the participant’s ID number, a column for the home address (Eircode), a column for the County (Dublin), and a column for the Country (Ireland). The CSV file was then added as a spatial layer into an open-source GIS application that supports viewing, editing, and analysis of geospatial data, Q(GIS). The identification of deprived and affluent areas was carried out using the *Pobal HP Deprivation Index (2011) spatial layer* at the level of “Small Areas” [34]. “Small Area” units usually comprise 80–100 households per unit and, therefore, offer a significantly better level of detail of the neighborhood in terms of spatial data [34]. The deprivation index data used were focused particularly on census data from 2011 (the most recently available during the recruitment process). The deprivation index range included being extremely disadvantaged (1), very disadvantaged (2), disadvantaged (3), marginally below average (4), marginally above average (5), affluent (6), very affluent (7), and extremely affluent (8). Due to the final sample size and in order to facilitate statistical analysis, the areas were then re-coded by collapsing responses 1–3 = high deprived; 4–5 = medium deprived; and 6–8 = low deprived.

### 2.3. Statistical Analysis

The data were analyzed using SPSS (macOS Mojave, version 26.0 (IBM, New York, NY, USA)). Associations between categorical variables (parental perceptions of neighborhood and deprivation index) were descriptively assessed using cross-tabulations and Chi-squared tests. For 3 × 2 contingency tables, Yates’ Continuity Correction was used to improve the Pearson Chi-square approximation. Adjusted binary logistic regression was performed on the parental perceptions of their local neighborhood variables and deprivation index, children’s dietary intake, active play, structured physical activity, and TV screen time. Models were adjusted for parents’ age, education, BMI, and household income. There was no evidence of collinearity (r < 0.7 and VIF < 2) in the adjusted models. Once the adjusted analysis had been performed, the usefulness of each model was assessed [35].

## 3. Results

### 3.1. Characteristics of Participants

An overview of participant characteristics is provided in Table 1. Boys and girls were almost equally represented (49.3%, *n* = 136 boys), and the mean age of children in this study was 4.42 (SD 2.67) years. Almost three-quarters (72.7%, *n* = 200) of children consumed at least one portion of fruit daily, and 40% consumed vegetables daily. The majority (70%, *n* = 194) of children consumed confectionary/SSBs once per week or more. Although over 50% (*n* = 152) of children spent more than 1 h watching television daily, over three-quarters (75.5%, *n* = 206) also spent more than 1 h engaged in active play, and 44% (*n* = 122) attended structured physical activity such as dance, football, and swimming classes.

The majority (88.4%, *n* = 244) of parents who participated in this study were mothers. Almost two-thirds (64.9%, *n* = 179) were in the age range of 30–39 years, and most were married or living with their partner (81.2%, *n* = 244) and were of Irish nationality (65.6%, *n* = 181). Over half the parents (60.9%, *n* = 168) were educated to higher degree levels either as undergraduate or postgraduate.

### 3.2. Neighborhood Deprivation Index and Parental Perceptions of Their Local Neighborhood

The distribution of households by deprivation index and parental perceptions of their neighborhood environment are presented in Table 2. Almost half (45.1%, *n* = 119) of the households were located in medium-deprived areas, almost one-third (31.8%, *n* = 84) were in low-deprived, and 23% (*n* = 73) were in high-deprived areas. Most (74.6%, *n* = 194) parents perceived their neighborhood as pleasant and safe for walking and cycling. They were satisfied with the number of physical activity facilities (50.4%, *n* = 130) and used them more than once per week (66.3%, *n* = 173). Parents also showed high satisfaction with the quality and availability of their local neighborhood food shops (grocery shops and supermarkets at 81.5%, *n* = 212, and 77.3%, *n* = 201, respectively) and restaurants (sit-in and takeaways) (48.8%, *n* = 127, and 50.4%, *n* = 138). In the bivariate analysis, five variables for parental perceptions and satisfaction were associated with deprivation index: walking/cycling is unsafe due to crime, pleasant for walking and cycling, satisfied with the number of physical activity facilities, usage of neighborhood facilities, and satisfied with neighborhood as a place to live/raise a child.

Table 3 shows parental perceptions and satisfaction with their neighborhood by deprivation index after adjusting for parent age, education, BMI, and household income. Of the twelve parental perception and satisfaction variables, only two made a significant contribution to the adjusted models. Parents who lived in high deprived areas were 15% more likely to perceive their neighborhood as unsafe for walking and cycling due to crime (OR 1.59, 95% CI 1.04–2.43, *p =* 0.031) when compared to parents who lived in low deprived areas. Parental satisfaction with their neighborhood as a place to live and raise a child was positively associated with low deprived neighborhoods (OR 1.28, 95% CI 1.02–1.59, *p =* 0.027).

### 3.3. Parents’ Satisfaction with Neighborhood Food Environment, Deprivation Index and Children’s Dietary Intake

No association between parental satisfaction with their local food environment and children’s intake of fruit and vegetables was observed. Three of the six neighborhood food environment satisfaction variables were associated with children’s intake of confectionary/SSBs after adjustment for parent education, age, BMI, and household income (Table 4). Children whose parents reported high satisfaction with the number of local sit-in and takeaway restaurants were 41% more likely to consume confectionary/SSBs weekly compared to children whose parents reported low satisfaction with the number of restaurants in their neighborhood (OR 1.41, 95% CI 1.08–1.81, *p =* 0.016). Similarly, high satisfaction with the number and quality of local food shops was positively associated with children’s weekly intake of confectionary/SSBs (OR 1.46, 95% CI 1.40–1.05, *p =* 0.020) and OR 1.43, 95% CI 1.05–1.94, *p =* 0.039, respectively). In the adjusted models, the neighborhood deprivation index was not associated with child dietary intake.

### 3.4. Parental Perceptions of Neighborhood Environment, Deprivation Index and Children’s Active Play, Structured Physical Activity, and TV Screen Time

Table 5 presents the adjusted analysis of parental perceptions of neighborhood deprivation index and children’s active play, structured physical activity, and TV screen time. Children whose parents showed higher satisfaction with their neighborhood as a place to live/raise a child were over 50% (OR1.41, 95% CI 1.01–1.98, *p* = 0.013) more likely to engage in a structured physical activity compared to children whose parents reported low satisfaction with their neighborhood as a place to live/raise a child. In contrast, children whose parents gave high scores for the perception of their neighborhood as unsafe for walking and cycling due to heavy traffic were less likely to engage in active play ≥ 1 h/d (OR 0.73, 95% CI 0.55–0.98, *p =* 0.041). Children’s active play for ≥ 1 h/d was positively associated with parents reported high use of parks, playgrounds, and/or swimming pools (OR 1.66, 95% CI 1.21–2.27, *p* = 0.004). Children’s television viewing was not associated with parents’ perceptions of the neighborhood physical activity environment after adjustment for confounding factors.

Neighborhood deprivation index was found to be associated with children’s structured physical activity in the adjusted model. Children who lived in a high deprived areas were 75% less likely to engage in a structured physical activity such as swimming, football, or dance compared to children from medium and low deprived neighborhoods (OR 0.35, 95% CI 0.17–0.72, *p =* 0.004) (Table 5).

## 4. Discussion

The present study aimed to explore the associations between deprivation index, pa- rental perceptions of their neighborhood food and physical activity environments, and children’s dietary intake, active play, structured physical activity, and TV screen time. Our results showed that most parents reported positive perceptions and high satisfaction with all aspects of the neighborhood environment such as it being pleasant and safe for walking and cycling, the number of physical activity facilities, and the quality and availability of their local restaurants and food shops (grocery shops and supermarkets). However, the neighborhood deprivation index was associated with some of these parental perceptions/satisfactions. For example, parents from highly deprived areas perceived their neighborhood as unsafe for walking and cycling due to crime, while parents from low deprived areas were highly satisfied with their neighborhood as a place to live and raise a child.

Indeed, deprived neighborhoods are normally characterized by a higher crime rate [36,37]. In Ireland, data from the latest census 2016 showed that 19% of people living in high deprived areas were “all the time” or “often” concerned about crime that could cause them physical harm, while this concern was only shared by 12% of people from low deprived areas [38]. Children living in high deprived areas have previously been shown to have a less supportive social environment for physical activity compared to children who live in low deprived areas [39]. Parental perceptions of neighborhood danger due to crime or even heavy traffic can affect their willingness to take the children to local physical activity facilities, parks, or playgrounds [40,41]. Conversely, parents that reside in affluent neighborhoods tend to develop and maintain a neighborhood social organization that promotes a safe, healthy, and positive environment for children [22,42].

In addition, parental perceptions and satisfaction with their neighborhood can influence how parents promote and limit their children’s interaction with the neighborhood food and physical activity facilities [42]. Our findings suggest that parents’ satisfaction with neighborhood restaurant and food outlets was positively associated with children’s intake of confectionary/SSBs. Children’s active play was negatively associated with parents’ perceptions of the neighborhood being unsafe for walking and cycling due to heavy traffic and positively associated with the usage of local physical activity facilities. Moreover, children’s structured physical activity was positively associated with parents’ satisfaction with the neighborhood as a place to live and raise a child and negatively associated with neighborhood high deprivation index.

An international study with 3670 children and their primary caregivers reported that children whose caregivers perceived high access to food outlets in their neighborhood were less likely to consume vegetables, fruits, and sugar-free breakfast cereals and more likely to eat takeaway/fast foods [15]. These results may suggest that there are aspects of children’s dietary intakes that may be influenced by parental perceptions of quality and availability of local food outlets. Indeed, availability and accessibility of takeaway/fast-food outlets and convenience stores have previously been shown to be associated with poorer quality dietary intake in children, while availability and accessibility of supermarkets have been associated with better diet quality [43]. In addition, parents’ perceptions of neighborhood food availability may be associated with home food availability and dietary intake [15].

Regarding children’s active play, results from international studies support our findings [10]. A study with 6 to 11 (*n* = 724) year old children reported that children whose parents perceived their neighborhood as safe for walking and cycling were more likely to use public recreation facilities such as parks and playgrounds [10]. Furthermore, parents’ satisfaction with quality and park design was also associated with park usage [44,45,46]. According to a systematic review of the literature, factors that may encourage park and playground usage include being close to home, providing a space that supports sport, playground equipment, and being safe and clean [44]. Moreover, many studies have supported the hypothesis that children who more frequently use parks/playgrounds have a lower BMI, less screen time, and greater overall physical activity [10,11,13,47,48,49,50]. In this context, addressing the structural (e.g., increasing the availability/accessibility of parks/playgrounds) and the social aspects of neighborhoods could be an important approach to increasing physical activity in pre-school children [9].

Our findings suggested an association between SES and children’s structured physical activity. Children’s participation in sports such as dance, swimming, and football was associated with parents’ satisfaction with neighborhood as a place to live and raise a child which, in turn, was associated with low deprivation index. In addition, children from high deprived areas were less likely to participate in these structured physical activities compared to children from low deprived areas. An Australian study with 402 parents of 5–17 year old children evaluated the barriers to children’s participation in organized sports. The authors reported that financial costs were strongly associated with children’s low participation in structured physical activity in lower income families [51]. A qualitative study with 113 children aged 10–11 years from 11 primary schools in the UK suggested that children from middle to high socioeconomic schools (SES) engaged more in structured activity such as after-school and weekend sports clubs while children from low SES schools engaged to a greater extent in active play. The authors also highlighted the financial cost as a barrier to children’s participation in structured physical activity [52]. Therefore, it is possible that our study results may be related to the financial costs of structured physical activity that may be problematic for families from high deprived areas [53].

Children living in high deprived areas have previously been shown to be at higher risk of developing obesity and having unhealthy eating habits and lower physical activity compared to children living in low deprived neighborhoods [22,26,37]. A nationally representative UK sample of 3717 children aged 3–7 years reported that children living in disadvantaged neighborhoods were at the greatest risk of overweight and central obesity [37]. Another study from the United States (US) with older children (*mean* age 10.9 years, SD 0.75) reported that children from affluent neighborhoods were more likely to eat healthily and spend less time on screen-related activities [36]. In contrast to these previous studies, our findings in an Irish urban/suburban setting did not observe a relationship between neighborhood deprivation index and children’s dietary intake, active play, and TV screen time after adjusting for confounding factors. The present study findings could be due to the age of the population; pre-school children have a low level of independence regarding their mobility in the neighborhood, food choice, and food purchase. Parents are the gatekeepers of children’s health behaviors; therefore, the home environment is an important setting in shaping children’s physical activity, food consumption, and weight status [16,54]. In addition, a strong association between children’s dietary intake, TV screen time, and aspects of the home environment, such as parental role modeling for diet and screen time, was observed in our previous studies [16,55].

### Study Strengths and Limitations

Before drawing conclusions, the strengths and limitations of this study must be considered. This is the first study in Ireland to provide insights into the relationship between the neighborhood environment and pre-school children’s dietary intakes, physical activity, and TV screen time behavior. Although the majority of participants lived in medium deprived areas, the deprivation extremes (high vs. low) were almost equally distributed, which allowed direct comparison between the two groups. The use of validated and standardized instruments allowed direct comparison with previous studies of children in the same age group.

The present study has limitations that also need to be considered. The study design is cross-sectional; therefore, it can only demonstrate associations between the variables investigated but cannot demonstrate cause and effect. The data collected were not nationally representative but derived from an urban/suburban setting. According to the most recent Irish Census (2016), 612,018 people of a total population of 4.995 million in Ireland speak a language other than English at home, and most non-Irish nationals live in Dublin City (91,876 of a population of 1.42 million) [56]. In the present study the number of potential non-English speakers was not recorded; therefore, we cannot compare data between English and non-English speaking populations. This is acknowledged as a limitation of the current study. Data on some potentially confounding factors such as participants’ use of food-delivery services, household car ownership, use of public transport, and information on the impact of weather on the frequency of playground usage were not collected. For example, greater access to transportation could increase participants’ choice of food outlets or parks/playgrounds outside their neighborhood environment. This may impact parental perceptions/use of their neighborhood food and physical activity facilities. Furthermore, the self-reported questionnaires used in this study may have introduced positive response bias among interested parents. Moreover, the assessment of lifestyle behaviors is known to be challenging, and parental recall of their child’s diet, physical activity, and TV screen time may be a further limiting factor.

## 5. Conclusions

The present study showed that neighborhood deprivation index was associated with parents’ perceptions of their neighborhood. Parents from high deprived areas perceived their neighborhood as unsafe for walking or cycling due to crime while parents from low deprived areas were satisfied with their neighborhood as a place to live in and raise a child. Therefore, special attention should be given to high deprived neighborhoods in terms of addressing structural and social factors that could provide an important approach to incentivize physical activity in young children from lower SES. In addition, outdoor active play and recreational facilities represent important opportunities for children to engage in physical activity. Our findings showed that children whose parents reported use of neighborhood physical activity facilities more frequently engaged in more active play. Parents, therefore, should be encouraged to spend more time with their children in parks/playgrounds and other safe outdoor places.

In addition, high parental satisfaction with the number of local restaurants, takeaway, and fast-food outlets was associated with higher children’s intake of confectionary/SSBs after adjusting for SES factors. Our findings suggest that parental perceptions of the neighborhood food environment can impact on children consumption of “unhealthy” foods. Such data are currently unavailable in Ireland, and the results of this study provide a unique understanding of the association between deprivation index, parents’ perception of their neighborhood, and indicators of healthy and unhealthy lifestyle in pre-school children. It is anticipated that findings from the present study could inform the development and implementation of intervention strategies in Ireland, aiming to improve local neighborhood environments to ensure availability and accessibility of healthier food options and physical activity facilities for this age group. Further examination of parental interaction with the neighborhood, including shopping behaviors, mobility, use of local food outlets, and physical activity facilities, should also be considered. Exploring parents’ views about their local neighborhood and focusing on highly deprived areas would help in better understanding the impact of neighborhood physical and social environment in low-income families.

## Figures and Tables

**Table 1 children-08-00959-t001:** Characteristics of the children and their parents.

Children’s Characteristics	Mean	SD
Age (*n* = 276)	4.42	2.67
Child z-BMI (*n* = 166)	0.78	0.96
Attending pre-school (*n* = 275)		
Full-time	97	35.3
Part-time	178	64.7
	*n*	%
Gender (*n* = 276)		
Male	136	49.3
Children’s lifestyle behaviors		
Fruit intake (*n* = 276)		
≥1 portion a day	200	72.7
<1 portion a day	75	27.3
Vegetable intake (*n =* 276)		
≥1 portion a day	110	40.1
<1 portion a day	164	59.9
Confectionary/SSB intake (*n* = 276)		
≥1 time per week	194	70.0
<1 time per week	83	30.0
Active play time (*n* = 276)		
≥1 h daily	206	75.5
<1 h daily	67	24.5
Attend structured physical activity (*n* = 276)	122	44.4
TV screen time (*n* = 276)		
≥1 h daily	152	55.5
<1 h daily	122	44.5
Parents’ Characteristics	*n*	%
Age (*n* = 276)		
20–29	35	12.7
30–39	179	64.9
≥40	62	22.5
Relationship with child (*n* = 276)		
Mother	244	88.4
Father	29	10.5
Other	3	1.1
Nationality (*n* = 276)		
Irish	181	65.6
Not Irish	95	34.4
Marital status (*n* = 275)		
Married or living together	244	81.2
Education level (*n* = 276)		
Undergraduate/post-graduate	168	60.9
Secondary school or less	108	39.1
Parental BMI (*n* = 254)		
Normal weight	227	82.2
Overweight/obese	27	9.8
Household income		
<40.000 €/p.a.	70	32.4
≥40.000 €/p.a.	146	67.6

**Table 2 children-08-00959-t002:** Parents’ perceptions of their neighborhoods by neighborhood deprivation index.

	Deprivation Index						
Parents’ Perceptions of Their Neighborhood	Total		High Deprived		Low Deprived		
	*n*	%	*n*	%	*n*	%	*p* ^1^
Walking/cycling is unsafe (heavy traffic) ^2^							
Disagree	158	61.00	32	52.5	57	68.7	0.208
Neither	45	17.40	10	16.4	13	15.7	
Agree	56	21.6	19	31.1	13	15.7	
Walking/cycling is unsafe (crime) ^2^							
Disagree	206	78.90	38	62.3	76	91.6	<0.001
Neither	36	13.80	13	21.3	4	4.8	
Agree	19	7.30	10	16.4	3	3.6	
Pleasant for walking and cycling ^2^							
Disagree	30	11.5	11	18.0	4	4.8	0.016
Neither	36	13.8	13	21.3	8	9.6	
Agree	194	74.6	37	60.7	71	85.5	
Satisfied with the number of physical activity facilities ^3^							
Dissatisfied	54	20.8	20	32.8	10	12.2	0.038
Neither	26	10.0	5	8.2	7	8.5	
Satisfied	180	69.2	36	59.0	65	79.3	
Usage of neighborhood facilities							
<1 time per week	88	33.7	32	52.5	19	22.6	0.001
≥1 time per week	173	66.3	29	47.5	65	77.4	
Satisfied with neighborhood as a place to live/raise a child ^3^							
Dissatisfied	63	24.4	15	24.6	6	7.2	0.001
Neither	65	25.2	12	19.7	7	8.7	
Satisfied	130	50.4	34	55.7	70	83.3	
Satisfied with the number of food shops ^3^							
Satisfied	201	77.3	52.00	85.2	63	75.9	0.151
Neither	26	10.0	4.00	6.6	12	10.0	
Dissatisfied	33	12.7	5.00	8.2	8	9.6	
Satisfied with the quality of food shops ^3^							
Dissatisfied	33	12.7	5	8.2	9	10.8	0.315
Neither	15	5.8	4	6.6	7	8.4	
Satisfied	212	81.5	52	85.2	67	80.7	
Satisfied with the number of restaurants ^3^							
Dissatisfied	63	24.4	14	23.3	14	16.9	0.250
Neither	65	25.2	14	23.3	25	30.1	
Satisfied	130	50.4	32	53.3	44	53.0	
Satisfied with the quality of restaurants ^3^							
Dissatisfied	61	23.5	16	26.2	13	15.7	0.300
Neither	72	27.7	16	26.2	23	27.7	
Satisfied	127	48.8	29	44.0	47	56.6	

^1^ *p* < 0.05 is significant. ^2^ Response options: strongly disagree (1); disagree (2); neither disagree nor agree (3); agree (4); and strongly agree (5) were re-coded as 1 and 2 = disagree; 3 = neither; and 4 and 5 = Agreed. ^3^ Response options: (1) strongly dissatisfied; (2) dissatisfied; (3) neither; (4) satisfied; and (5) strongly satisfied were re-coded as 1 and 3 = dissatisfied; 3 = neither; and 4 and 5 = satisfied.

**Table 3 children-08-00959-t003:** Associations between parents’ perceptions of their neighborhood and deprivation index.

	Deprivation Index			
Parents’ Perception of the Neighborhood ^2^	High Deprived		Low Deprived	
	OR (95% CI)	*p* ^1^	OR (95% CI)	*p* ^1^
Walking/cycling is unsafe (heavy traffic)	0.84(0.59–1.21)	0.374	1.04(0.78–1.39)	0.744
Walking/cycling is unsafe (crime)	1.59(1.04–2.43)	0.031	0.73(0.48–1.10)	0.141
Pleasant for walking and cycling	0.74(0.48–1.15)	0.186	0.97(0.85–1.11)	0.735
Satisfied with the number of physical activity facilities	0.79(0.54–1.15)	0.231	1.04(0.73–1.47)	0.825
Usage of neighborhood physical activity facilities	0.86(0.70–1.07)	0.186	1.25(0.98–1.58)	0.067
Satisfied with neighborhood as a place to live/raise a child	0.83(0.66–1.04)	0.115	1.28(1.02–1.59)	0.027
Satisfied with the number of restaurants	1.23(0.73–2.07)	0.425	0.97(0.63–1.50)	0.917
Satisfied with the quality of restaurants	0.84(0.48–1.46)	0.548	1.25(0.79–1.99)	0.329
Satisfied with the number of food shop	1.18(0.64–2.16)	0.587	0.80(0.48–1.33)	0.404
Satisfied with the quality of food shop	1.52(0.81–2.84)	0.183	0.99(0.59–1.66)	0.974

CI: Confidence Interval; OR: Odds Ratio; ^1^ *p* < 0.05 is significant. Values are OR that were obtained from the binary logistic regression models adjusted for parental education, BMI, age, and household income. ^2^ High scores represent positive parental perceptions and satisfaction with their local neighborhood.

**Table 4 children-08-00959-t004:** Logistic regression models for parents’ satisfaction with neighborhood food environment, deprivation index, and children’s fruit, vegetable, and confectionary/SSB intakes.

	Fruit		Vegetables		Confectionary/SSBs	
Characteristic	OR (95% CI)	*p* ^1^	OR (95% CI)	*p* ^1^	OR (95% CI)	*p* ^1^
Satisfied with the quality of restaurants	1.05(0.77–1.44)	0.729	1.12(0.50–1.48)	0.417	1.27(0.95–1.70)	0.098
Satisfied with the number of restaurants	1.03(0.76–1.39)	0.822	0.96(0.7–1.25)	0.790	1.41(1.08–1.81)	0.016
Satisfied with the number of food shops	0.84(0.59–1.19)	0.342	0.91(0.68–1.23)	0.917	1.46(1.40–1.05)	0.020
Satisfied with the quality of food shops	0.88(0.62–1.22)	0.469	0.91(0.67–1.24)	0.917	1.43(1.05–1.94)	0.039
Deprivation index						
High deprived	1.26(0.55–2.90)	0.575	0.55(0.23–1.31)	0.180	0.99(0.41–2.34)	0.981
Medium deprived ^(Ref.)^						
Low deprived	1.41(0.64–3.11)	0.387	0.95(0.44–1.86)	0.900	1.2(0.61–2.52)	0.575

CI: confidence interval; OR: odds ratio; Ref.: Reference group. ^1^ *p* < 0.05 is significant. Values are OR that were obtained from the binary logistic regression models adjusted for parental education, BMI, age, and household income.

**Table 5 children-08-00959-t005:** Associations between parents’ perception of their neighbourhood, deprivation index, and children’s active play, structured physical activity and TV screen time.

	Active Play		Structured Physical Activity		TV Screen Time	
Characteristic	OR (95% CI)	*p* ^1^	OR (95% CI)	*p* ^1^	OR (95% CI)	*p* ^1^
Walking/cycling is unsafe (heavy traffic)	0.73(0.55–0.98)	0.041	1.03(0.80–1.33)	0.787	0.86(0.66–1.12)	0.285
Walking/cycling is unsafe (crime)	0.91(0.64–1.30)	0.618	0.85(0.65–1.14)	0.290	0.79(0.57–1.09)	0.163
Pleasant for walking and cycling	1.26(0.87–1.82)	0.221	1.05(0.77–1.43)	0.746	1.10(0.79–1.52)	0.561
Satisfied with the number of physical activity facilities	1.13(0.82–1.55)	0.443	1.15(0.89–1.48)	0.281	0.81(0.62–1.06)	0.059
Use of neighborhood physical activity facilities	1.66(1.21–2.27)	0.004	1.19(0.96–1.48)	0.091	0.84(0.67–1.05)	0.094
Satisfied with neighborhood as a place to live/raise a child	0.94(0.78–1.13)	0.078	1.41(1.01–1.98)	0.013	1.04(0.88–1.23)	0.597
Deprivation index						
High deprived	0.65(0.27–1.56)	0.338	0.35(0.16–0.76)	0.008	0.89(0.38–2.04)	0.787
Medium deprived ^(Ref.)^						
Low deprived	0.54(0.25–1.17)	0.120	0.61(0.33–1.13)	0.119	0.613(0.31–1.20)	0.155

CI: confidence interval; OR: odds ratio; Ref.: Reference group. ^1^ *p* < 0.05 is significant. Values are OR that were obtained from the final binary logistic regression models adjusted for parental education, BMI, age, and household income.

## Data Availability

The data presented in this study are available on request from the corresponding author. The data are not publicly available due to ethical considerations.

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
