# Peer review of "Associations between Neighborhood Deprivation Index, Parent Perceptions and Preschooler Lifestyle Behaviors"

_children, 2021, doi:10.3390/children8110959_

Round 1

Reviewer 1 Report

This study is well designed.

More details are needed for the GIS evaluation.

Reviewer 2 Report

Thank you for the opportunity to review this manuscript focused on a timely topic. The aims are clearly described and the study design and methods are adequate.  However, there are a few issues the auhors would need to address.

1.- Methods. Participants and study procedures

The authors report that The Pre-schoolers Health Study, a cross-sectional study, comprised 332 child-parent dyads,  n = 670 children were elegible, but in the case of siblings only one child per parent was elegible. A total of 276 parents completed the questionnaire.

Does this mean that the present study was part of The Pre-schoolers Health Study?

Was it an additional study on the same sample, with a different number of respondents?

Please, clarify.

2. Data collection instruments

The authors report how they used the information collected, but they do not inform how did they ask about diet. 

Did you use a single food frequency question for fruits and  one item for vegetables? In that case, potatoes were included?
Which were the answer categories?

The same regarding confectionaery and SSBs.

3.- Neighbourhood deprivation index. 

In the introduction you refer to a deprivation index. Did you used that deprivation index? You should add suitable references and provide some information in the methods section. 

4.- Discussion

Findings regarding diet need further discussion. Additional conditioning factors that could influence food choices?

Reviewer 3 Report

You have prepared a very nicely written and presented paper on your study. Childhood obesity is still an important public health problem and this study does add some additional useful information to support interventions with young children and their families. I have only a few comments and suggestions and most related to how you discuss your findings. * I note your study did not include non-English speakers but it would be useful to know if the there is a high non-English speaking population in Dublin and therefore what the impact of this might be. Please note this and discuss any limitations that might be relevant. * lines 125-126 - 'parent self-reported height and weight of child' - we can guess what you mean but better to make it clearer * Line 180 - what does the (42) mean? * line 270 - typo - confounding *Discussion: You have presented the study so well and then the discussion is a bit mess, especially at the start. I suggest you start with the positive relationships you found and summarise these in the first paragraph. Leave reference to other studies till AFTER you have summarised your KEY findings, ALSO adding why you believe this information is important/useful for the field. THEN go on to say what findings were unexpected, or not similar to what is already in the literature. Eg. Where you talk about the the lack of observed relationship between neighborhood deprivation index and children's health related behaviour. It would also be better to try to explain this finding itself - that is, perhaps deprivation index is not so important for pre-school children - suggest why. ONLY then refer to other information which is clearly in other papers you have written from this study (one problem of salami slicing your findings!) that might help explain it. Are these other factors better predictors of children's health behaviours? Are these other factors you mention related to the deprivation index? Are they therefore stronger predictors or mediators? You mention 'takeaways and fast food outlets' in the discussion but there is no mention of 'takeways' (presumably takeway/fast food shops) in the methods or discussion so it wasn't clear that this was specifically asked about. If not, I would remove reference to 'takeways', otherwise include it in the methods or results. I don't think your findings strongly suggest that satisfaction with local shops is leading to increased confectionary/SSB consumption - it is a bit of a leap to make this assumption. You do not have enough data to support this so please consider removing or reducing this. I think the parental perceptions of neighborhood crime/safety will affect the parents' willingness to take the children to local physical activity facilities for outdoor play - these are pre-school children so I don't think the parents will be 'encouraging them' to to use local facilities (lines 318/319) I suggest amending the wording. While using neighborhood physical activity facilities supports active play, parents satisfaction with facilities appeared to have some relation to use of facilities even if not statistically significant. I think it would be worth mentioning this as it supports the theory of why use is increased. I would also suggest that this may mean that the quality of the local parks/playgrounds may be a factor in use, rather than just availability or accessibility so suggest you consider including this.

Reviewer 4 Report

This is an interesting study. I have few comments to the current presentation.

Abstract

  1. I reckon that line 17 “, using validated and standardized questionnaires” is abundant.
  2. Spell out the SSB.

Introduction

  1. How is your primary measure associated with children overweight?
  2. The introduction regarding the relationship between children overweight status and deprivation index is out scope of the study since your study did not use quantitative measurements for body composition. Perhaps focused on parental deprivation index to children’s physical activities, TV screen time, and dietary intake is more appropriate.
  3. How is the criminal rate affecting children’s playing time in Ireland? Does it affect preschool children play just in public facilities or private institute? It seems the participants all from kindergarten should have time to play around during the daily routine.
  4. Line 83-85. Another paragraph should establish for this concept.

Method

  1. Your study period was across winter, spring, and summer. Does seasonal change affect the parents’ responses? How this matter affects your statistical analysis?
  2. Line 165, item 8 is missing.

Result

  1. Please unify the symbol of p value in the tables.

Discussion

  1. As mentioned above, how your study determine overweight/obesity? Do your study collect BMI or any other physical measures?
  2. High deprived parents considered crime as a factor to affect engagement in walking and cycling. How is the criminal situation between the high deprivation and low deprivation parents lived? It is interesting to see “satisfied with neighbourhood as a place to live/raise a child” is main concern in low deprived parents but not in high deprived parents.
  3. Any geographical or cultural difference among the 25 pre-schools? Such environmental conditions, industrial area etc. may be potential factors for the outcome measures.
  4. Practical application of the study findings to Ireland society is far more important than just laundry the results of the study.

Round 2

Reviewer 4 Report

The authors have done a great work for the resubmission. The current form of manuscript is suitable for publication in Children.